# Cefiderocol Resistance in *Pseudomonas aeruginosa* ST175: A Case Report with Genomic Analysis

**DOI:** 10.3390/antibiotics14111162

**Published:** 2025-11-17

**Authors:** Rosario Fernández Fernández, Alberto Badillo Carrasco, Natalia Chueca Porcuna, Antonio Martínez Cabezas, Manuel Colmenero Ruiz

**Affiliations:** 1Intensive Medicine, Hospital San Cecilio, 18016 Granada, Spain; 2Microbiology Unit, Hospital San Cecilio, 18016 Granada, Spain; alberto.badillo.sspa@juntadeandalucia.es (A.B.C.); natalia.chueca.sspa@juntadeandalucia.es (N.C.P.); 3Institute of Biosanitary Research (ibs.GRANADA), Hospital San Cecilio, 18016 Granada, Spain; 4Spanish Consortium for Research on Infectious Diseases (CIBERINFEC), 28029 Madrid, Spain; 5Intensive Care Medicine Service, Hospital San Cecilio, 18016 Granada, Spain; manuel.colmenero.sspa@juntadeandalucia.es; 6Department of Medicine, Facultad Medicina, Universidad Granada, 18012 Granada, Spain

**Keywords:** cefiderocol, *Pseudomonas aeruginosa* ST175, IMP carbapenemase, multidrug resistance, ventilator-associated pneumonia, case report

## Abstract

Background: Cefiderocol, a novel siderophore cephalosporin, has shown promising activity against multidrug-resistant (MDR) Gram-negative pathogens, including metallo-β-lactamase (MBL) producers. However, emerging reports suggest rapid resistance development under selective pressure, particularly in epidemic *Pseudomonas aeruginosa* lineages. Case presentation: We describe a 50-year-old man with a history of aortic dissection who developed ventilator-associated pneumonia due to *P. aeruginosa* ST175 carrying an IMP-type carbapenemase. Initial treatment with ceftazidime/avibactam plus aztreonam was followed by cefiderocol monotherapy. Although the patient showed early improvement, clinical relapse occurred within days, with subsequent isolation of cefiderocol-resistant *P. aeruginosa*. Whole-genome sequencing revealed high-impact mutations in iron acquisition systems (including fptA and pvdE) and multiple resistance determinants (oprD, efflux pump regulators, ampD/ampR), disrupting siderophore-mediated uptake and favoring multidrug resistance. Despite rescue therapy with ceftazidime/avibactam plus aztreonam and adjunctive agents, the patient ultimately died from septic shock and multiorgan failure. Discussion: This case highlights the ability of high-risk *P. aeruginosa* clones to rapidly adapt to cefiderocol, driven by combined alterations in iron metabolism and classical resistance pathways. The development of resistance within less than one week of exposure underscores the risk of using cefiderocol as monotherapy in infections caused by epidemic MDR clones. Conclusions: Cefiderocol resistance can emerge rapidly in vivo during treatment of infections caused by *P. aeruginosa* ST175. Continuous surveillance, molecular characterization of resistance mechanisms, and cautious use of cefiderocol—preferably in combination regimens—are warranted to preserve its clinical utility.

## 1. Introduction

Siderophores are small molecules produced by bacteria and fungi to facilitate iron acquisition, an essential nutrient that acts as a cofactor in multiple critical cellular processes [1,2]. The ability to produce and utilize siderophores represents a key microbial adaptation. Stable ferri–siderophore complexes are transported into the periplasm through TonB-dependent transporters (TBDTs) located in the outer membrane. This uptake process requires the TonB inner membrane complex (TonB, ExbB, ExbD), which couples proton motive force to energize TBDTs and mediate conformational changes that allow translocation of siderophore–iron complexes into the periplasm [3,4,5,6,7]. Once inside, iron may be released or the intact complex further transported into the cytoplasm.

*Pseudomonas aeruginosa* produces the siderophores pyoverdine and pyochelin and can also utilize xenosiderophores produced by other microorganisms [8]. Several TBDTs, including FiuA and FpvB, participate in ferrichrome uptake, and loss of both transporters markedly impairs bacterial growth [9]. High-risk clones such as ST175 are widely disseminated in Europe, particularly in Spain and France, and have been predominant among extensively drug-resistant (XDR) isolates in Spain for more than a decade, accounting for 40–85% of XDR cases [10,11,12].

Cefiderocol is a novel siderophore cephalosporin specifically designed to combat multidrug-resistant Gram-negative bacteria [13]. Its structure incorporates a catechol-type siderophore moiety and cephalosporin backbone, enabling active uptake via iron transport systems in addition to passive diffusion through porins [14]. These features confer activity against non-fermenters such as *P*. *aeruginosa* and *Acinetobacter baumannii*, as well as *Enterobacterales*. However, the increasing global prevalence of carbapenem resistance in *P*. *aeruginosa*, *Acinetobacter* spp., and *Enterobacterales* represents an urgent public health threat [15]. Although new agents (e.g., ceftolozane–tazobactam, ceftazidime–avibactam, imipenem–relebactam, meropenem–vaborbactam) have expanded treatment options, resistance mechanisms beyond β-lactamases—such as porin loss and efflux pump upregulation—remain a major challenge [16,17,18].

Among the most pressing concerns is the accelerated dissemination of metallo-β-lactamase (MBL)-producing pathogens [19]. MBLs comprise a subclass of carbapenemases capable of hydrolyzing a broad range of β-lactam antibiotics, with the most clinically consequential variants belonging to the New Delhi MBL (NDM), Verona integron-encoded MBL (VIM), and imipenemase (IMP) families [19].

In a 2025 study published in Chemical Science, investigators examined the variable activity of cefiderocol against MBL-producing pathogens [20]. They showed that NDM-1 and NDM-5 hydrolyze cefiderocol efficiently, whereas VIM-2 and IMP-1 exhibit minimal cefiderocol turnover. The hydrolysis product of cefiderocol showed inhibitory effects on VIM-2 and IMP-1, but not on NDM variants [21]. Although MBL families degrade cefiderocol to some extent, only NDM enzymes do so at a rate sufficient to drive clinically meaningful resistance. Supporting this mechanistic explanation, isogenic strains expressing NDM-1 exhibit markedly higher cefiderocol minimum inhibitory concentrations than those expressing IMP-1. These results explain clinical failures observed in NDM-producing infections and suggest that cefiderocol might exhibit greater efficacy against VIM-producing or IMP-producing strains.

Here, we report the case of a patient infected with *P*. *aeruginosa* ST175 who developed cefiderocol resistance during treatment. This case illustrates the capacity of epidemic high-risk clones to rapidly adapt to novel antimicrobials and underscores the need for vigilance when using cefiderocol in clinical practice. There is a tendency to assume uniform susceptibility—or resistance—to cefiderocol across all MBL-producing organisms.

## 2. Case Report

50-year-old man with a history of Stanford type A aortic dissection (previously treated with supracoronary Dacron graft replacement and carotid–carotid bypass) was admitted for elective repair of a residual Stanford type B dissection and underwent thoracic endovascular aortic repair (TEVAR).

Immediately after surgery, he developed distributive shock requiring norepinephrine, vasopressin, and corticosteroids. He remained intubated and febrile, with elevated inflammatory markers. Empirical therapy with ceftazidime/avibactam and daptomycin with meropenem was started; ceftaroline was later added after blood cultures yielded *S*. *haemolyticus* (1/2 bottles) de-escalation of previous antibiotics. Transesophageal echocardiography ruled out endocarditis.

On day 2, meropenem was started for a CT scan revealed a pulmonary infiltrate consistent with pneumonia was confirmed on CT (early VAP). Despite persistent fever, inflammatory markers decreased. By day 12, the patient deteriorated with new respiratory failure. Bronchoalveolar lavage (BAL) and urine cultures grew *P*. *aeruginosa*, and amikacin was added.

On day 14, the isolate was confirmed as IMP-carbapenemase-producing *P*. *aeruginosa*. Therapy was changed to ceftazidime/avibactam plus aztreonam in dual infusion. Recurrent *S*. *haemolyticus* bacteremia required reintroduction of daptomycin.

By day 19, *P*. *aeruginosa* was again isolated from BAL, and cefiderocol monotherapy was initiated. After an initial response, fever and inflammatory markers reappeared within 72 h.

PET-CT on day 30 excluded prosthetic infection, blood cultures were negative, and BAL cultures cleared. Cefiderocol stopped on day 35 after 15 days of therapy.

On day 36, fever and inflammatory markers recurred, and BAL cultures grew *P*. *aeruginosa* now resistant to cefiderocol but susceptible to ceftazidima/avibactam + aztreonam. Therapy was re-initiated, with clinical improvement. However, on day 44, the patient developed catheter-associated *E*. *faecalis* bacteremia, with persistence of IMP-producing *P*. *aeruginosa* in BAL (cefiderocol-resistant, ceftazidima/avibactam + aztreonam-susceptible). Daptomycin, ceftaroline ceftazidima/avibactam + aztreonam and corticosteroids were administered, but septic shock ensued.

On day 51, the patient developed severe adult respiratory distress syndrome (ARDS) and multiorgan dysfunction. Despite initiation of intravenous colistin and isavuconazole, his condition went worse, dying from multiple organ failure.

Table 1 summarizes sequential susceptibility results for *P. aeruginosa* isolates collected across different dates and times, showing a clear trend toward increased resistance to most agents, while colistin and aztreonam/avibactam remained susceptible; all susceptibility testing was performed using the MicroScan WalkAway automated system.

## 3. Genomic Analysis

To investigate the molecular mechanisms underlying cefiderocol resistance development, whole-genome sequencing libraries were prepared and run on Illumina HiSeq, with sequences deposited in BV-BRC under Genome IDs IMP_924606_CefS (cefiderocol-susceptible index), IMP_cefiR_946746, IMP_cefiR_964382, and IMP_cefiR_969748 (cefiderocol-resistant).

Comparative genomic analysis revealed multiple mutations potentially associated with cefiderocol resistance development. The most significant findings included frameshift mutations in genes encoding TonB-dependent transporters involved in siderophore uptake, particularly fptA (ferric-pyochelin uptake transporter) and pvdE (pyoverdine uptake system). Additional mutations were identified in genes related to heme utilization pathways and iron homeostasis, which may contribute to altered siderophore-mediated drug uptake.

Table 2 summarizes the acquired resistance determinants and key chromosomal mutations identified in the sequenced IMP-producing *P. aeruginosa* isolates. The detected genes and mutations are consistent with multidrug resistance and may contribute to reduced cefiderocol susceptibility.

The genomic analysis revealed that cefiderocol resistance emerged through a combination of mechanisms affecting both drug uptake and efflux. Notably, the disruption of siderophore uptake systems (fptA and pvdE mutations) likely reduced the active transport of cefiderocol into the bacterial cell, while maintaining the IMP-23 metallo-β-lactamase provided additional enzymatic inactivation of the drug.

## 4. Discussion

This case report describes the rapid emergence of cefiderocol resistance in a clinical isolate of *P*. *aeruginosa* ST175 during treatment, highlighting several important clinical and microbiological considerations. The development of resistance within 10 days of therapy initiation underscores the adaptive capacity of high-risk clones and the potential for treatment failure when using cefiderocol as monotherapy against MBL-producing organisms.

The genomic analysis provides insights into the molecular mechanisms underlying cefiderocol resistance development. The frameshift mutations in fptA and pvdE genes, which encode key components of the ferric-pyochelin and pyoverdine uptake systems, respectively, represent a novel resistance mechanism specific to siderophore-based antibiotics. These mutations effectively block the “Trojan horse” strategy employed by cefiderocol, forcing the drug to rely solely on passive diffusion through porins for cellular entry.

The presence of IMP-23 metallo-β-lactamase in the initial isolate likely provided a baseline level of cefiderocol hydrolysis, as demonstrated in previous biochemical studies [20]. However, the relatively low initial MIC (2 mg/L) suggests that enzymatic hydrolysis alone was insufficient to confer high-level resistance. The subsequent emergence of transport-related mutations represents an additional layer of resistance that, combined with the existing MBL activity, resulted in clinically significant resistance levels.

The rapid development of resistance observed in this case aligns with recent reports of cefiderocol heteroresistance in carbapenem-resistant Gram-negative pathogens [17]. Heteroresistance may facilitate the selection of resistant subpopulations under antibiotic pressure, particularly when drug concentrations fluctuate below optimal levels at the infection site.

While cefiderocol has been positioned as one of the few active agents against carbapenemase-producing *P*. *aeruginosa*, our case highlights a worrisome phenomenon: rapid in vivo resistance development during therapy. The ST-175 isolate carried the metallo-β-lactamase IMP-23, and phenotypic testing confirmed resistance to cefiderocol within the first weeks of treatment. This observation is particularly concerning given that cefiderocol was expected to retain activity against metallo-β-lactamase-producing strains and suggests that the acquisition of specific resistance determinants in this high-risk background may compromise its utility.

The epidemiological significance of this case extends beyond the individual patient outcome. *P*. *aeruginosa* ST175 represents one of the most successful high-risk clones globally, with widespread dissemination in healthcare settings across Europe [10,11,12], particularly in Spain where ST175 remains the dominant lineage in hospitals. The capacity of this clone to rapidly develop cefiderocol resistance through multiple molecular mechanisms raises concerns the need for continuous surveillance of cefiderocol susceptibility, molecular characterization of resistance mechanisms, and careful stewardship of novel β-lactam–β-lactamase inhibitor combinations in order to preserve their activity.

## 5. Conclusions

This case report demonstrates the rapid emergence of cefiderocol resistance in *P*. *aeruginosa* ST175 through mutations affecting siderophore uptake systems. The findings highlight the need for careful monitoring of cefiderocol susceptibility during treatment and consideration of combination therapy, particularly for infections caused by MBL-producing high-risk clones. Continued surveillance of resistance mechanisms and clinical outcomes will be essential to optimize the use of this important therapeutic agent.

## Figures and Tables

**Table 1 antibiotics-14-01162-t001:** Suscetibility results for *P. aeruginosa.*

Antibiotic	17 April 2025	17 April 2025	4 May 2025	7 May 2025	11 May 2025	27 May 2025
**Ceftazidime**	>32 (R)	>32 (R)	>32 (R)	>32 (R)	>32 (R)	>64 (R)
**Cefepime**	>8 (R)	>8 (R)	>8 (R)	>8 (R)	>32 (R)	>32 (R)
**Colistin**	≤2 (S)	1 (S)	≤2 (S)	≤2 (S)	≤2 (S)S	≤2 (S)S
**Aztreonam**	16 (I)	16 (I)	>16 (R)	>16 (R)	>16 (R)	>16 (R)
**Aztreonam/Avibactam**	≤4 (S)	≤4 (S)	S	S	S	2 (S)
**Piperacillin/Tazo**	16 (R)	16 (R)	>16 (R)	>16 (R)	-	>64 (R)
**Meropenem**	>32 (R)	>32 (R)	>32 (R)	>32 (R)	>32 (R)	>32 (R)
**Amikacin**	>16 (R)	>16 (R)	>16 (R)	>16 (R)	>16 (R)	>16 (R)
**Ceftolozane/Tazo**	>4 (R)	>4 (R)	>4 (R)	>4 (R)	>8 (R)	>8 (R)
**Cefiderocol**	**2 (S)**	**S**	**3 (R)**	**R**	**R**	**R**

(S = Susceptible, I = Increment, R = Resistant. MIC values are in μg/mL where specified by context, and interpretation is based on EUCAST guidelines used in the source documentation).

**Table 2 antibiotics-14-01162-t002:** Acquired resistance determinants and mutations potentially affecting cefiderocol susceptibility in *P*. *aeruginosa.*

**Acquired Resistance Determinants**		
**Gene**	**Antibiotic class affected**	**Enzyme/protein type**
GES-5	Carbapenems, cephalosporins, penams	GES-type β-lactamase
IMP-23	Carbapenems, cephalosporins, cephamycins, penams, penems	IMP-type metallo-β-lactamase
**Mutations potentially affecting cefiderocol activity**		
**Gene**	**Mutation type**	**Functional consequence**
fptA	Frameshift (Val66fs)	Disrupts ferric-pyochelin uptake
pvdE	Frameshift (Ala347fs)	Disrupts pyoverdine uptake
nirL	Start lost (His173Arg)	Affects heme d_1_ biosynthesis
tpsA [1]	Premature stop (Ser3039*)	Impairs heme utilization pathways
tpsA [4]	Frameshift (Gly951fs)	Same as above
cdrA	Frameshift (Ala456fs)	Same as above
ftpC	Frameshift (Gly497fs)	Same as above
ShlA/HecA/FhaA	Frameshift (Gly815fs)	Same as above
oprD	15 mutations (1 high-impact)	Linked to carbapenem resistance
mexA, mexC, mexE, mexS, mexX, mexY	Various mutations	Affect efflux pump systems (antibiotic export)
gyrA, parC, parS	Various mutations	Impair DNA replication processes
nalC	Various mutations	Involved in quinolone resistance
ampD, ampR	Various mutations	Regulate β-lactamase production

## Data Availability

Data supporting the findings of this study are available from the corresponding author upon reasonable request.

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
