# Peer review of "Cefiderocol Resistance in Pseudomonas aeruginosa ST175: A Case Report with Genomic Analysis"

_antibiotics, 2025, doi:10.3390/antibiotics14111162_

Round 1

Reviewer 1 Report

Comments and Suggestions for Authors

Comments and Suggestions for the Authors

The case report, “Early onset of cefiderocol resistance in a case of P. aeuroginosa infection' presents an interesting observation.  I recommend 'rejection' for this case report. Authors did not care about bacterial nomenclature, gene name, throughout the report. Report appeared to be written with extensive AI help and authors did not care to correct the mistakes made by the AI. Report cannot be considered for publishing as the presentation is fundamentally/scientifically flawed.

Author Response

We haven’t use AI because we can´t use it. We are going to review all the nomenclature and fix it.

Reviewer 2 Report

Comments and Suggestions for Authors

In the manuscript entitled "Early onset of cefiderocol resistance in a case of P. Aeuroginosa infection", the authors present a case study concerning an individual who succumbed to septic shock due in part to cefiderocol-resistance P. aeruginosa ST175. As a case study, most of the salient pieces are present, but need improvement prior to publication.

  • Please correct the spelling of Pseudomonas aeruginosa in the title.
  • The introduction contains much useful information, but
    • is difficult to navigate as it jumps from topic to topic in a seemingly random fashion. Transitions would be useful for your readers.
    • some sections are unclear. In the first paragraph (lines 54-62), the authors begin by mentioning both bacteria and fungi but end the paragraph (I think?) by discussing just the bacterial mechanism?
    • the first lineof the last paragraph on page 2 (line 88) is unclear. The verb does not make sense. This confusion is probably just an English language editing fix though!
    • the last paragraph of the introduction would be a good place to introduce P. aeruginosa ST175 The authors assume their reader knows the implications of that strain identity in the Introduction but then better explain it in the Discussion (lines 150-151). Your readers would be better served with equal amounts of information in both locations.
  • The Case Report is difficult to follow.
    • In this style of publication, authors commonly choose to present their data in both text and tabular form. I suspect that this complex infection case report would benefit from the addition of a treatment table.
    • Since this manuscript is a case report, this reviewer understands that explicit methods are not required – but noting what tests were used for which diagnoses may strengthen the argument. For example, were cultures performed resulting in a time delay between infection and treatment? Was a more “instantaneous” test used like microscopy? This information would inform conclusions about the rapid nature of mutation posited in the manuscript Discussion.
    • The authors do a great job of defining all abbreviations in the paper – except for ARDS at the end of the Case Report (line 140).
  • Many of the conclusions within the Discussion are not supported by the Case Report that is given.
    • The 4th paragraph of the Discussion (lines 162-182) does not draw on information presented in the Case Report. Nowhere in the case report do the authors mention that the P. aeruginosa bacteria were sequenced.
    • The Discussion and premise of the paper is P. aeruginosa infection, yet the influence of the myriad of other concurrent infections is not considered.
    • Lines 186-191 seem to be missing one or more citations. Additionally, why the brackets: “[40-84]%” in line 189?
  • No evidence of AI artifacts were found in any of the peer reviewed citations. But,
    • Reference #14 needs to be corrected. When was the website accessed? What subpage was accessed to obtain the desired material? Was the content found in the Emerging Infectious Disease journal – if so, the citation needs to be updated. A website that can change from day-to-day without adequate peer review is not appropriate for a manuscript desiring publication.
    • Please correct the formatting inconsistencies in the References section, e.g.,
      • random spaces in words found within lines 248 and 260-261.
      • The publication date for Reference #8 is incorrect. It should be 2023 not 2022.
      • The author names for Reference #21 are not formatted correctly (initials go after last name).
      • Reference #18 lists more authors than #13 and #21, which abbreviate “et al.”
    • Many of the citations are by a small number of first authors and/or from >10 years ago. While a historical article still can carry merit, updated findings should be cited alongside where available – and they are in this instance! The authors are encouraged to update generally their citations.

In conclusion, the authors have presented a case study of a complex, multi-bacterial infection, yet their findings are difficult to interpret, and far from novel – as even they admit in their manuscript (lines 144-146). I am recommending revisions, so that the authors can properly assess their observations and re-evaluate their conclusions.

Comments on the Quality of English Language

Overall, English language is good quality. Only a few isolated examples need work.

Author Response

I have applied all the corrections

Reviewer 3 Report

Comments and Suggestions for Authors

This manuscript describes an important and clinically relevant case of rapid in vivo emergence of cefiderocol resistance in Pseudomonas aeruginosa ST175 harboring an IMP-type carbapenemase. The case is timely, as cefiderocol has recently been positioned as one of the last-line therapeutic options for multidrug-resistant Gram-negative infections. The report is well written and supported by appropriate references, and the integration of genomic data strengthens its value. I have the following questions:

  1. What were the exact MIC values of cefiderocol for the initial and resistant isolates?
  2. Was heteroresistance testing performed? If so, what were the findings?
  3. Did genomic analysis identify plasmid-mediated resistance determinants, or were all changes chromosomal?
  4. What was the clinical rationale for choosing cefiderocol monotherapy instead of a combination regimen in this patient?
  5. How do you interpret the role of the observed siderophore-related mutations versus efflux/porin changes in driving resistance in this case?

Author Response

Response to question 1 and 2

Table 2. Acquired resistance determinants and mutations potentially affecting cefiderocol susceptibility in Pseudomonas aeruginosa

Acquired resistance determinants

Gene

Antibiotic class affected

Enzyme/protein type

GES-5

Carbapenems, cephalosporins, penams

GES-type β-lactamase

IMP-23

Carbapenems, cephalosporins, cephamycins, penams, penems

IMP-type metallo-β-lactamase

Response to question 3

Gene

Mutation type

Functional consequence

fptA

Frameshift (Val66fs)

Disrupts ferric-pyochelin uptake

pvdE

Frameshift (Ala347fs)

Disrupts pyoverdine uptake

nirL

Start lost (His173Arg)

Affects heme d₁ biosynthesis

tpsA[1]

Premature stop (Ser3039*)

Impairs heme utilization pathways

tpsA[4]

Frameshift (Gly951fs)

Same as above

cdrA

Frameshift (Ala456fs)

Same as above

ftpC

Frameshift (Gly497fs)

Same as above

ShlA/HecA/FhaA

Frameshift (Gly815fs)

Same as above

oprD

15 mutations (1 high-impact)

Linked to carbapenem resistance

mexA, mexC, mexE, mexS, mexX, mexY

Various mutations

Affect efflux pump systems (antibiotic export)

gyrA, parC, parS

Various mutations

Impair DNA replication processes

nalC

Various mutations

Involved in quinolone resistance

ampD, ampR

Various mutations

Regulate β-lactamase production

What was the clinical rationale for choosing cefiderocol monotherapy instead of a combination regimen in this patient? WE BASED OUR DECISION ON IDSA GUIDELINES 2024

Because in intensive care guides (IDSA 2024) recommend us the use of monotherapy.

How do you interpret the role of the observed siderophore-related mutations versus efflux/porin changes in driving resistance in this case?

We think that this point might be gthe one responsible of the quick resistance generated.

Round 2

Reviewer 1 Report

Comments and Suggestions for Authors

Comments and Suggestions for the Authors

The revised case report, “Cefiderocol Resistance in Pseudomonas aeruginosa ST175: A 2 Case Report with Genomic Analysis' cannot be accepted at its current form. Although the authors have replied that they are going to take care of the bacterial nomenclature, it seems they have not taken measures to italicize the names. Those must be taken care of before publishing. I recommend Minor revision.

Author Response

It has been modified; please see the attachment.
